# Reactive Oxygen Species in Acute Lymphoblastic Leukaemia: Reducing Radicals to Refine Responses

**DOI:** 10.3390/antiox10101616

**Published:** 2021-10-14

**Authors:** Abdul Mannan, Zacary P. Germon, Janis Chamberlain, Jonathan R. Sillar, Brett Nixon, Matthew D. Dun

**Affiliations:** 1Cancer Signalling Research Group, School of Biomedical Sciences and Pharmacy, College of Health, Medicine and Wellbeing, University of Newcastle, Callaghan, NSW 2308, Australia; Abdul.Mannan@newcastle.edu.au (A.M.); Zacary.Germon@uon.edu.au (Z.P.G.); Janis.Chamberlain@health.nsw.gov.au (J.C.); Jonathan.Sillar@uon.edu.au (J.R.S.); 2Hunter Medical Research Institute, New Lambton Heights, NSW 2305, Australia; Brett.Nixon@newcastle.edu.au; 3John Hunter Children’s Hospital, New Lambton Heights, NSW 2308, Australia; 4Calvary Mater Hospital, Waratah, NSW 2298, Australia; 5Reproductive Science Group, College of Engineering, Science and Environment, University of Newcastle, Callaghan, NSW 2308, Australia

**Keywords:** acute lymphoblastic leukaemia, reactive oxygen species, oxidative stress, NADPH oxidases, antioxidants, redox homeostasis, second messenger signalling, oxidative DNA damage, resistance, oncogenic signalling, cysteine oxidation, kinase, phosphatase

## Abstract

Acute lymphoblastic leukaemia (ALL) is the most common cancer diagnosed in children and adolescents. Approximately 70% of patients survive >5-years following diagnosis, however, for those that fail upfront therapies, survival is poor. Reactive oxygen species (ROS) are elevated in a range of cancers and are emerging as significant contributors to the leukaemogenesis of ALL. ROS modulate the function of signalling proteins through oxidation of cysteine residues, as well as promote genomic instability by damaging DNA, to promote chemotherapy resistance. Current therapeutic approaches exploit the pro-oxidant intracellular environment of malignant B and T lymphoblasts to cause irreversible DNA damage and cell death, however these strategies impact normal haematopoiesis and lead to long lasting side-effects. Therapies suppressing ROS production, especially those targeting ROS producing enzymes such as the NADPH oxidases (NOXs), are emerging alternatives to treat cancers and may be exploited to improve the ALL treatment. Here, we discuss the roles that ROS play in normal haematopoiesis and in ALL. We explore the molecular mechanisms underpinning overproduction of ROS in ALL, and their roles in disease progression and drug resistance. Finally, we examine strategies to target ROS production, with a specific focus on the NOX enzymes, to improve the treatment of ALL.

## 1. Introduction

Acute lymphoblastic leukaemia (ALL) is a heterogeneous malignancy of immature B or T lymphoblasts, which rapidly proliferate in the bone marrow, blood, and some extra-medullary sites such as the spleen and lymph nodes [1]. ALL is the most common form of childhood cancer [2], and although only 20% of ALL diagnoses are diagnosed in adults, four out of five deaths from ALL are in this age group [3]. The overall survival (OS) of children diagnosed with ALL has dramatically improved over the last 40 years. Indeed, the development of multidrug treatment regimens including vincristine [1,4], corticosteroids [5], and asparaginase [6], with most regimens adding an anthracycline [7] (usually doxorubicin or daunorubicin) has reduced treatment resistance, and led to remission rates of greater than 80% [4]. The backbone of ALL treatment is similar in adults; however, they have worse outcomes due to both higher risk disease features at diagnosis and more toxicities associated with therapies [1,4,8]. However, in both populations the early failure of upfront therapies has devastating consequences, with a median 5-year OS of 21% for children [8] and 2% for adults who relapse within their first year of diagnosis [9]. These concerning data highlight the need to continually develop treatments for ALL patients at diagnosis and at disease progression.

A handful of risk factors are associated with ALL including, prenatal exposure to X-rays, postnatal exposure to high doses of radiation and previous treatments with chemotherapy [10], with a genetic predisposition seen in a subset of ALL cases. These include rare genetic and familial cancer syndromes, DNA polymorphisms in non-coding genes and numerous germline variants in coding genes [11]. Chromosomal abnormalities are also common in ALL and include gain or loss of chromosomal content (aneuploidy) and chromosomal rearrangements. Typical chromosomal translocations include, t(9;22) [BCR/ABL1], t(12;21) [ETV6/RUNX1], t(1;19) [TCF3/PBX1], and Mixed-lineage leukaemia (MLL)-rearrangements [1]. Overall consequences of chromosomal abnormalities are the loss of tumour suppressor genes or production of chimeric proteins that dysregulate many cellular processes particularly those that underpin cellular development, differentiation, multiplications, and cell cycle regulation [4]. Recurring somatic and occasionally germline mutations in transcription factors (*IKZF1, STAT5*) [12,13], tumour suppressors *TP53* (including germline variants), *CDKN2A* [14,15], and signalling pathways genes such as NOTCH1 [16], PI3K/Akt (*FLT3*, *PTEN, PTPN11*) [17,18,19,20], JAK/STAT (*CLRF2*, *IL7R*, *JAK1*, *JAK3*) [21,22] and Ras (*BCR/ABL, NRAS, KRAS*) [23,24] drive malignant transformation of immature lymphocytes and perturb the function of the body’s immune system. Recurring mutations in signalling genes are strongly associated with pathways that underpin the increased production of reactive oxygen species (ROS); oxidative radicals that induce DNA damage leading to genomic instability and promote leukaemogenesis [25]. However, the roles of ROS in redox signalling and genome instability in ALL remains enigmatic and infantine. In this review, we summarise the known roles that ROS fulfil in the dysfunction of ALL blasts and discuss therapeutic interventions with particular attention given to strategies that reduce ROS production used in combination with established standard-of-care chemotherapies and targeted therapies.

## 2. Reactive Oxygen Species

ROS are a heterogenous group of small molecules, ions and radicals derived from oxygen molecules that share the property of being highly reactive due to the presence of unpaired electrons. Some of the common examples of ROS include superoxide anion (O_2_^∙−^), nitric oxide (NO^∙−^), hydroxyl radical (HO^∙−^), hydrogen peroxide (H_2_O_2_), ozone (O_3_), and lipid peroxides [26]. The main free-radical, superoxide anion, and the non-free radical, hydrogen peroxide, are the two most important ROS, with each playing multiple roles in the regulation of cellular signalling and biological processes [25]. The role of ROS in normal and cancer-associated signalling have been studied extensively over the last couple of decades [27], revealing multiple mechanisms underpinning their cancer associated effects, including the modulation of cell signalling [28], oxidative DNA damage [29], promoting genomic instability [30], and lipid peroxidation [31]. It follows that cancer stemness and self-renewal [32,33], cancer progression, resistance to standard-or-care treatment [34] and relapse are all associated with the levels of ROS production in cancer [29,35]. Other than cancer, ROS also play deleterious roles in other diseases such as neurodegeneration [36], fertility [37], ageing [38], diabetes [39], and hypertension [39].

## 3. Sources of Reactive Oxygen Species

The nicotinamide adenine dinucleotide phosphate (NADPH) oxidases (NOXs) and mitochondrial respiratory chain complexes are the two main sources of intracellular ROS production [26]. The NOX family of enzymes are transmembrane proteins embedded in the plasma membrane, predominantly in immune cells and produce ROS as part of the phagosome [40]. Neutrophils, monocytes, and macrophages lyse invading pathogens using ROS produced by a NOX2 initiated respiratory burst [41,42,43]. Multiple stimuli, including bacterial lipopolysaccharide (LPS), tumour necrosis factor-alpha (TNF), and interleukin (IL-1), can activate NOXs [44]. However, the expression of the NOX family of enzymes is not limited to immune cells; they are also found in various tissues including the intestine [45], vascular smooth muscles [46], lung epithelium [47], kidneys [48], spleen, uterus [49], endothelial cells [50] testis, lymph nodes [51], and the brain to aid in angiogenesis [52] and are pivotal in regulating a multitude of normal cellular processes. NOX-driven ROS production is implicit in redox regulated post-translational modifications (PTMs) as is frequently seen in cysteine containing enzymes. Such oxidatively driven PTMs result in the modulation of enzyme structure and folding with downstream consequences for their cellular signalling activity and hence ability to influence gene expression and cellular differentiation [44,53].

The catalytic subunits of human NOX enzyme complexes consist of p22*^phox^* (*CYBA*), NOX1 (*NOX1*), NOX2, commonly referred to as gp91*^phox^* (*CYBB*), NOX3 (*NOX3*), NOX4 (*NOX4*), NOX5 (*NOX5*), Dual oxidase 1 (*DUOX1*), and DUOX2 (*DUOX2*). NOX2 was the first NOX isoform characterised and it catalyses the production of superoxide on plasma membranes by transferring one electron from the intracellular NADPH to extracellular or luminal oxygen. The catalytic subunits of NADPH oxidases (NOX1-5, DUOX1-2) also require interaction with regulatory subunits to modulate their activity. NOX2, for example, has six regulatory subunits: two membrane-bound p22*^phox^* and Ras-related protein 1 (Rap1), and four cytoplasmic p40*^phox^* (*NCF4*), p47*^phox^* (*NCF1*), p67*^phox^* (*NCF2*) and the Ras-related C3 botulinum toxin substrate (*RAC1* and *RAC2*) proteins. The activation of NOX1-4 requires heterodimerisation with p22*^phox^*, with NOX5 and DUOX1-2 activated independent of p22*^phox^* but requires calcium. While all other NOX isoforms catalyse the production of superoxides, NOX4 and DUOX1-2 produce hydrogen peroxides through the peroxidase-like domain [44]. Therefore, the differential expression of these isoforms can determine the dominant ROS produced in a specific tissue and/or organelle.

Mitochondria are the other major source of ROS production. Indeed, superoxide was first discovered from the mitochondria, wherein it is produced by the electron transport chain complexes located within the inner mitochondrial membrane [54,55]. Specifically, during oxidative phosphorylation electron leakage from the respiratory chain enzyme complexes I and III release superoxide anion [56]. Superoxide anions are subsequently converted into the more stable hydroxyl radical (HO^−^) and hydrogen peroxide (H_2_O_2_) [57]. Endoplasmic reticulum and peroxisomes can also contribute to cellular H_2_O_2_. The relative stability of H_2_O_2_ ensures a longer half-life than superoxide, and the chemical has the added properties of being able to radially diffuse across cell membranes via aquaporins (peroxiporins), act as a secondary messenger in cellular signalling [58], and react with cysteines leading to temporary or permanent redox PTM of signalling proteins [59]. Despite increased stability, H_2_O_2_ still can damage lipids and proteins especially in the presence of iron, via a phenomenon called the Fenton reaction [59].

Mammalian cells have evolved complex antioxidant systems that can counter the toxic effects of elevated H_2_O_2_ and hence cellular oxidative stress (Figure 1) (reviewed in [60]). These antioxidant systems are mainly divided into enzymatic and non-enzymatic processes that are crucial in maintaining a balance between oxidative stress and essential redox signalling, often referred to as redox homeostasis. Enzymatic antioxidants include superoxide dismutase (SOD), catalase (CAT), and glutathione peroxidase (GPx), thioredoxin (TRX), peroxiredoxin (PRX) and glutathione transferase (GST) [61]. The spontaneous or enzymatic dismutation of superoxide by SOD produces H_2_O_2_, oxygen (O_2_) and water (H_2_O); however, H_2_O_2_ can be also be reduced to hydroxyl radicals (HO) [62]. There are three different isoforms of SOD; manganese (Mn)-SOD (SOD2) which is mainly located in the mitochondria, SOD1 in the cytoplasm and SOD3 in the extracellular space [58]. Oxidative stresses such as ionisation radiation induces the expression of SOD2 to balance increased oxidative stress observed within the mitochondrial space [63]. Importantly, knock-out of SOD2 is neonatally lethal in mice, a phenotype that results from excessive oxidative DNA damage and biochemical aberrations within the mitochondria, thus highlighting the essential role of SOD2 in mitochondrial function and redox homeostasis [64].

Cysteinyl residues within certain proteins such as in peroxiredoxins and selenocysteinyl residues in GPx (myeloperoxidase—*MPO*, eosinophil peroxidase—*EPX*, lactoperoxidase—*LPO*) react with the H_2_O_2_ and lipid hydroperoxides, and as such, form the basis of the mammalian antioxidant system [65]. GPx reduces H_2_O_2_ and lipid peroxides to their respective alcohols using low molecular weight thiols such as GSH; one of the most abundant non-protein thiols found ubiquitously in mammalian cells. GPx catalyses the oxidation of reduced GSH and converts it into a disulphide-oxidised form (GSSG), thereby also converting H_2_O_2_ into H_2_O and O_2_. GPx, TRX and PRX are other thiol-containing enzymes involved in the clearance of H_2_O_2_ in cells. These antioxidant systems require NADPH to maintain the pool of reduced substrates for the reduction of H_2_O_2_. For example, GSSG reductase uses NADPH to convert GSSG into its reduced form, GSH [61]. The first-rate limiting enzyme of the pentose phosphate pathway, glucose-6-phosphate dehydrogenase, generates intracellular NADPH by reducing NADP^+^. Therefore, glucose-6-phosphate dehydrogenase is essential to maintain GSH and is regarded as a regulatory antioxidant enzyme. The GST family of enzymes mainly inactivate secondary metabolites, including epoxides and hydroperoxides. Vitamin C, vitamin E, glutathione, carotenoids (β-Carotene) and flavonoids are among the more common of the non-enzymatic antioxidants. The antioxidant activity of these compounds rests with non-enzymatic mechanisms such as the donation of electrons (vitamin E (α-tocopherol), GSH), scavenging of O_2_ radicals (vitamin A), chelation of metal ions and trapping of free radicals [61].

As summarised, ROS are produced continuously within eukaryotic cells owing to constant metabolic and biochemical responses. Under normal physiological conditions, cells maintain a dynamic equilibrium between the production and clearance of ROS utilising the plethora of antioxidant systems described above. However, multiple external and internal factors, including ionising radiation, xenobiotic chemicals including anticancer drugs, and pathogens, can disrupt this equilibrium resulting in higher production and/or reduced clearance of ROS with a net shift towards increased oxidative stress levels. Owing to the role of ROS in activating multiple oncogenic signalling pathways and in promoting oxidative DNA damage, overproduction of ROS can result in the initiation and progression of multiple diseases, including ALL.

## 4. The Essential Role of ROS in the Maintenance of Haemopoietic Stem Cells (HSCs), and Innate and Adaptive Immunity

Haemopoietic stem cells (HSCs) are specialised cells residing within the complex environment of bone marrow endosteal region, called niches. HSCs can self-renew to maintain their pool and differentiate into each of the different blood cell types [25]. Bone marrow niches have a relatively low oxygen tension, thereby creating a hypoxic environment favouring the quiescent state of HSCs. Most HSCs are in a state of quiescence and produce low levels of adenosine triphosphates (ATPs) via anaerobic glycolysis to fulfil their energy demands [66]. Self-renewal and quiescence of HSCs is maintained in part by hypoxia-inducible factor 1 alpha (*HIF1A*) [67], the master regulator of hypoxia-inducible gene expression in response to low oxygen. Knockdown of *HIF1A* in HSCs reduces quiescence and self-renewal ability and increases ROS production [68,69].

In addition to HIF signalling, ataxia-telangiectasia mutation (ATM), forkhead box class O transcription factor (FoxO), Akt, and mTOR signalling pathways are also crucial in the regulation of redox homeostasis and HSCs quiescence (reviewed in [60]). FoxOs are transcriptional factors that act downstream to PTEN/PI3K/Akt pathway, considered essential for the self-renewal capacity of HSCs. Loss of *FoxO3a* in HSCs abrogates their long-term self-renewal ability, accompanied by increased phosphorylation of p38 MAPK, and elevation of ROS production [70]. Similarly, conditional deletion of *FoxO1/3/4* in adult HSCs reduces their long-term repopulating activity with a commensurate increase in cell cycling and apoptosis mediated through elevated levels of ROS. The ability of the antioxidant N-acetyl-L-cysteine (NAC) to reverse this defective phenotype of *FoxO*-deficient HSCs [71] highlights the role of ROS in HSC pathophysiology.

Upstream of FoxO, the PI3K/Akt pathway is one of the most upregulated pathways in cancer, including ALL [21,72]. The activation of PI3K/Akt promotes cell cycle progression and mitochondrial oxidative phosphorylation which is associated with increased ROS production in cancer [73]. Akt directly activates NOX to potentiate ROS production through the phosphorylation of p47*^phox^* [74]. Conditional deletion of *PTEN*, the negative regulator of PI3K/Akt signalling, enhances cell cycle progression, proliferation, differentiation, and reduces self-renewal of HSCs [75]. Similarly, chronic activation of Akt in HSCs increases proliferation and loss of HSCs self-renewal [76]. However, *Akt1/2*-deficient HSCs produce very little ROS and harbour very poor long-term transplantation efficacy due to persistent quiescence at the G0 phase of the cell cycle. Pharmacological increase of ROS rescues differentiation defects of *Akt1/2*-deficient HSCs [77].

The mammalian target of rapamycin (mTOR) is another downstream regulator of PI3K/Akt pathways and promotes mitochondrial oxidative phosphorylation and ROS production. The conditional deletion of tuberous sclerosis complex (*Tsc1*), the negative regulator of mTOR, drives cell cycle activity in murine HSCs through increased mitochondrial biogenesis and elevated ROS production, thereby impairing self-renewal. Rapamycin, an inhibitor of mTOR, rescued the effects of *Tsc1* deletion in mouse HSCs [78].

Ataxia-telangiectasia mutated (ATM, a member of the PIKK (PI3K-like protein kinase) family, is a key stress response protein heavily involved in DNA damage repair and through yet an unknow mechanism regulates ROS. Genetic deletion of (*Atm*^−^^/^^−^) increased ROS levels and reduced self-renewal ability of HSCs, rescued by treatment with the antioxidant NAC [79]. Redox PTMs in ATM increase NADPH production via the pentose phosphate shunt, with NADPH required for the reduction of oxidized glutathione and thioredoxin [80] (Figure 1), a possible mechanism of elevated ROS in ATM-deficient HSCs.

Janus kinases (JAKs) and Signal transducer and activator of transcription (STATs) proteins are important in maintenance of HSC self-renewal and stemness. *Jak2* knock-out (*Jak2*^−^^/^^−^) leads to foetal death through perturbations in erythropoiesis, while conditional loss of *Jak1* impairs self-renewal disrupting haematopoiesis [81]. *Jak2*^−^*^/^*^−^ increased ROS levels and p38 MAPK phosphorylation with the concomitant loss of HSCs self-renewal and bone marrow failure in mice [82]. Similarly, STAT3/STAT5, critical haematopoietic transcription factors are activated by phosphorylation by JAK, are implicated in the self-renewal capacity of HSCs [81]. Given JAK phosphorylates and activates STAT, which is linked to ROS production in leukaemia [30,83] (discussed in Section 6), it is interesting that *Jak2*^−^*^/^*^−^ knockout increased ROS levels in HSCs. It is quite possible that in HSCs, STAT activity promotes expression of antioxidant proteins, while aberrant DNA methylation in ALL leads to hypermethylation of antioxidant and tumour suppressor genes [84]. Importantly, the pathways involved in the self-renewal HSCs regulate the levels of ROS, emphasising the importance of redox homeostasis in haematopoiesis. In addition, multiple other proteins and pathways, such as tumour suppressors p53 (*TP53*) and retinoblastoma protein (RB), glycogen synthase kinase-3 (GSK-3), cyclin dependent kinases (CDK), and BCL2 all play an important role in self-renewal, maintaining quiescence, proliferation, differentiation and survival of HSCs mainly through the modulation of mitochondrial ROS [85]. ROS, therefore, plays an essential role in the quiescence, self-renewal, and long-term survival of HSCs, and hence, elevated, and sustained ROS exposure is likely to contribute to cell cycle progression, DNA damage and the initiation and progression of ALL.

ROS also plays an important role in both innate and adaptive immune responses [86]. Chronic granulomatous disease (CGD), a primary immunodeficiency condition in which individuals are highly susceptible to infection, is a classic example of the pivotal role that ROS plays in innate immunity. Mutations in NOX2 in neutrophils, impair phagocytosis as the respiratory burst is ineffective and hence these cells are unable to kill the invading pathogens [87]. In addition to NOX2, mitochondrial ROS has also been implicated in phagocytic bacterial lysis [88,89]. Bacterial activation of toll-like receptors (TLR1, TLR2 and TLR4) in macrophages increases mitochondrial ROS production and mitochondrial recruitment to the phagosome [89]. In addition, infection-induced endoplasmic reticulum stress increases mitochondrial ROS delivered to bacteria-containing phagosomes via mitochondria-derived vesicles (MDVs) [90].

ROS also plays important roles in the activation, proliferation and maturation of B and T cells, key to adaptive immunity [91,92]. Both B and T cells express NOX2 [93,94,95], albeit at reduced levels compared to neutrophils and macrophages. Activated T cells directly contact and stimulate the respiratory burst of neutrophils [96], aiding in innate immunity. T-cell receptor (TCR) ligation triggers the production of NOX-derived ROS. Loss of NOX2 not only reduces ROS production but also alters cytokine production. In human T cells, the TCR stimulates ROS production through the activation of the NOX family member DUOX1, rescued by *DUOX1* knockdown [97]. In addition to TCR stimulated ROS production in T cells, activation of CD28 which is necessary for T cell activation leads to ROS production and NF-kB induced interleukin-2 (IL-2) expression [98,99]. Mitochondrial ROS has also been shown to play an important function in T cell adaptive immune responses. Mitochondrial ROS induced by TCR-initiated calcium influx activates Nuclear factor of activated T cells (NFAT) to drive T cell activation and IL-2 production to drive antigen specific expansion in vivo and in vitro [92]. ROS are also involved the activation of B-cell receptors (BCR). High levels of ROS oxidatively inhibit phosphatases to activate BCR signalling further amplified through concomitant activation of Spleen tyrosine kinase (SYK) [100]. A recent study suggests that at the earlier stages of B cell activation, NOX2 is the main source of ROS, supplemented by mitochondrial ROS during immune response [101]. In addition, ROS has also been reported to increase the binding activity of PAX5, a transcription factor essential for B cell maturation. While molecular loss of *Pax5* reduces B cell differentiation [102], alterations to *PAX5* are seen in B-ALL [103].

## 5. Redox Dysregulation in ALL

Multiple lines of evidence support the notion that ROS play a role in the leukaemogenesis of ALL. ALL associated somatic heterogeneity and chromosomal translocations trigger constitutive activation of various oncogenic kinases, well-established contributors to increased ROS production in leukaemia [25]. Multiple pathways/mechanisms are involved in the generation of increased ROS in ALL (Figure 1). Some of the better characterised of these proteins/pathways are discussed below.

### 5.1. ETV6/RUNX1 Fusions

Hallmark ALL associated chromosomal abnormalities drive malignant transformation of HSCs and progenitor cells and are associated with increased ROS production. The common t(12;21)(p13;q22) chromosomal translocations occurs in 25% of paediatric B-cell precursor ALL (BCP-ALL) and result in a chimeric *ETV6/RUNX1* fusion gene [104]. *ETV6/RUNX1* translocation has been reported prenatally to initiate a pre-leukaemic state [105]. This “first-hit” incidence of *ETV6/RUNX1* fusion does not lead to malignant transformation; however, it can impair the function of wild type ETV6 and RUNX1, leading to cell cycle dysregulation [106]. Transgenic mouse models of ETV6/RUNX1 expressing CD19^+^ B cells, showed increased ROS and increased DNA damage [107]. Although the exact mechanism by which ETV6/RUNX1 increases ROS are not known, ETV6/RUNX1 binds with the promoter region of erythropoietin receptor (EPOR) to drive gene expression. Binding of EPO ligand with the EPOR in turn increases pre-B cell survival mediated by JAK2/STAT5 activity, which upregulates the antiapoptotic protein BCL-XL. Interestingly, NOX2-derived ROS in turn increases EPO signalling, suggestive of a positive feedback mechanism [108]. STAT5 binds and activates Rac1, promoting NOX mediated ROS production, to increase DNA damage in AML cell lines [30], a possible mechanism by which ETV6/RUNX1 induces ROS and garnishes the secondary hit necessary for preleukemic clones to propagate mutations that favour the development of fully transformed ALL.

### 5.2. BCR/ABL Oncogene

An alternative t(9;22)(q34;q11) translocation gives rise to the Philadelphia chromosome (Ph), which is present in approximately 3% of paediatric, and in 25% of adult B-ALL (Ph^+^ B-ALL) [109]. This translocation produces the Breakpoint cluster-Abelson (BCR/ABL) fusion protein, in which the tyrosine kinase domain of ABL1 becomes constitutively active. Interestingly, BCR/ABL has also been reported in healthy individuals [110], identifying this event as a putative first hit, but can transform cells by itself. Accordingly, BCR/ABL fusion protein activates multiple ROS producing signalling pathways, including PI3K/Akt/mTOR and JAK/STAT [111] (Figure 1). PI3K/Akt in turn increases ROS production through increased mitochondrial activity and NOX activation, [73] generating a positive feedback loop. BCR/ABL driven ROS [112], induces DNA damage and drug resistance [113]. High-level of ROS production in leukaemic blasts harbouring recurring BCR/ABL oncogenes is regulated by Rac independent activation of NOX4. BCR/ABL driven ROS can also oxidise and inhibit the tumour suppressor serine threonine phosphatase PP1α leading to unfettered PI3K/Akt signalling and increased expression of proteins involved in cell survival [114]. The NOX and flavoprotein inhibitor diphenyleneiodonium (DPI), and tyrosine kinase inhibitors, Imatinib and Nilotinib, have both been shown to reduce ROS in preclinical models of BCR/ABL driven leukaemia [114,115]. Conversely, BCR/ABL transgenic mice express activated Rac3 in primary precursor B lymphoblasts, with molecular inhibition of *Rac3* increasing survival of mice harbouring BCR/ABL+ leukaemia [116]. As Rac3 plays an important role in the activation of NOX1-3 [117], it is likely that BCR/ABL mutations increase Rac3-NOX mediated ROS production in ALL.

Not only does BCR/ABL influence redox signalling, but it also enhances the ability of cells to survive DNA damage by increasing expression of RAD51 and decreasing expression DNA protein kinase- DNA-PK (*PRKDC)*, key elements of double-strand break (DSB) repair, homologous recombination repair (HRR) and non-homologous end-joining (NHEJ), respectively [118,119,120,121]. In cases of adult acute myeloid leukaemias (AML) driven by constitutive activation of the FMS-like tyrosine kinase 3 (FLT3) present in 30–35% of cases [122]; Leukaemias well characterised by redox dysfunction [25], increased activity of the NHEJ DSBs repair pathway is also seen and regulated by the phosphorylation of S2612 DNA-PK [123].

### 5.3. Cytokine Receptor-like Factor 2 (CRLF2)

Approximately 20% of B-ALL patients harbour an activated kinase gene expression profile resembling Ph^+^ B-ALL but are negative for the BCR/ABL fusion [109]. Cytokine receptor-like factor 2 (*CRLF2*) encodes the receptor for thymic stromal lymphopoietin (TSLP), which activates cellular signalling on binding, required for normal lymphocyte function [124]. *CRLF2* rearrangements lead to the overexpression of *CRLF2* mRNA and protein [125] and are common in patients diagnosed with ‘Ph-like’ B-ALL [126]. Increased minimal residual disease (MRD) is frequently seen in Ph-like ALL patients following upfront standard induction chemotherapies, giving rise to disease relapse and poor overall survival [127]. Activation of CRLF2 stimulates PI3K/Akt/mTOR and JAK/STAT signalling to deregulate lymphoid progenitor differentiation, increases metabolism and, drives ROS production and associated genomic instability [82,128] (Figure 1). These studies suggest that CRLF2 induced ROS production can positively regulate its levels by constitutive activation of JAK/STAT signalling. Interestingly, the majority of Down syndrome (DS) patients who develop ALL (62%) harbour increased CRLF2 expression, with genomic signatures enriched for increased DNA damage [129]. Approximately 50% of Ph-like B-ALL patients with *CRLF2*-rearranged also harbour *JAK1* or *JAK2* mutations [125,130]. Overexpression of *Crlf2* and mutant *Jak2* drives factor-independent cell growth of murine BaF3 pro-B cells [129]. Furthermore, a recurring gain-of-function mutation in *CRLF2* (F232C) seen in ALL patients, promotes constitutive dimerisation of the receptor, which may be induced by ROS, leading to cytokine independent activation and cell survival [131].

### 5.4. Interleukin-7 Receptor α (IL7R)

The Interleukin-7 receptor α (*IL7R*) is required for normal lymphoid development. Somatic gain-of-function mutations in *IL7R* are seen in paediatric B and T -ALL. A S185C substitution in the extracellular domain or in-frame insertions and deletions in the transmembrane domain constitutively activates the IL7R [132]. *IL7R* mutations are dominantly linked with the aberrant expression of CRLF2, where mutant IL7R forms a functional receptor with CRLF2 for TSLP. Hence, IL7R signalling drives ROS production and redox dysfunction through the activation of PI3K/Akt/mTOR, JAK/STAT and ERK/MAPK pathways, all of which are implicated in ALL disease progression and treatment resistance [133,134,135].

### 5.5. Transcription Factors PU.1 (SPI1) and Spi-B (SPIB)

The E26-transformation-specific (ETS) transcription factors PU.1 (*SPI1*) and Spi-B (*SPIB*) are important tumour suppressor genes that regulate B cell development and function [136,137]. Conditional deletion of *Spi1* and *Spi1B* in pre-B-cells impairs B cell differentiation and drives ALL development through ROS production and acquisition of secondary driver mutations in *JAK1*, *JAK3* or *IKZF3* (IKAROS Family Zinc Finger 3). Acquired *JAK* mutations can further promote leukaemia progression through hyperactivation of JAK/STAT signalling and further increased ROS production driving DNA damage [138]. Conversely, the JAK inhibitor, ruxolitinib, delayed leukaemia onset and suppressed both ROS production and ROS associated gene expression signatures [139]. Decreased levels of PU.1 increase HSCs cell cycle progression through the activation of the MAPK pathway [140]. MAPK increases the activity of Activator protein 1 (AP-1), a transcription factor driving expression of the NOX activating subunit p22*^phox^* creating a feed-forward loop [141].

### 5.6. Neurogenic Locus Notch Homolog Protein 1 (NOTCH1)

NOTCH1 signalling is the most (70–80%) deregulated pathway in T-ALL [15]. NOTCH1 is a cell membrane receptor that plays a key role in the proliferation, differentiation, and activation of T cells. Mutations and constitutive activation of NOTCH1 in T-ALL promotes ROS production and PI3K/Akt signalling pathways indirectly through the regulation of C-myc [142], which in turn induces DNA damage [143] (Figure 1). PTEN, an upstream negative regulator of PI3K/Akt/mTOR suppresses and hence decreases ROS production. Importantly, ROS induce redox PTMs and inactivation of PTEN’s tyrosine phosphatase activity leading to sustained activation of PI3K/Akt/mTOR and thus ROS production through a positive feedback mechanism in T-ALL (Figure 1). Furthermore, Casein kinase 2 (CK2), which positively regulates the PI3K/Akt/mTOR pathway, is also elevated in NOTCH1 mutant T-ALL, further increasing ROS production and hyperactivation of PI3K/Akt/mTOR signalling. It follows that combinations of the NAC antioxidant and the CK2 inhibitor tetrabromobenzotriazole (TBB) can synergistically ablate NOTCH1 driven T-ALL survival [144].

### 5.7. Ras GTPases (N- and KRAS)

Activating mutations in either *N*- or *KRAS* are reported in 15–20% the paediatric ALL [145,146,147], and are enriched in relapsed and chemo resistant ALL patients [148]. Increased expression or activity of RAS guanyl-releasing protein 1(RASGRP1), a positive regulator of RAS/Raf/MEK pathway also occurs in ALL [149,150]. RAS proteins function as molecular switches that cycle between active and inactive states based upon the binding of guanosine triphosphate (GTP) and guanosine diphosphate (GDP), respectively. Normally, ligand mediated RTK activation is required to stimulate RAS and initiate a cascade of Raf/MEK signalling responsible for regulating multiple cellular functions including protein trafficking, ROS production and cellular proliferation [28,151]. However, mutations in codons 12, 13 and 61 impair the GTPase activity, constitutively activating RAS proteins and hence the downstream Raf/MEK pathway driving tumour growth [152]. Through the stimulation of NOX, constitutively active N-Ras^G12D^ and H-Ras^G12V^ upregulate ROS production promoting growth-factor independent proliferation of human CD34^+^ haematopoietic cells. Mechanistically, K-RAS activates p38 MAPK initiating a cascade of PDPK1/PKCδ/ p47*^phox^*/NOX1 signalling for the generation of ROS and cellular transformation (Figure 1). Given the role of RAS in NOX-induced ROS and recurring *RAS* lesions [24,145,146,147], it is very likely that constitutively active RAS drives ROS production in ALL, and as such patients may benefit from therapies targeting this ROS production pathway (discussed in Section 6). Indeed, MEK inhibition, a downstream target of RAS, synergises with prednisolone in the treatment of both RAS-mutant and wildtype ALL [153], potentially underpinned by reduced ROS production following RAS inhibition.

### 5.8. Rho-Family GTPases

The increased expression and activation of Rho GTPase family proteins RAC1-3 is common in ALL [116,154,155]. Like Ras, Rac GTPases also cyclically switch between GTP-bound active and GDP-bound inactive states. While Rac1 regulated the chemotactic response of ALL cells to chemokine SDF-1α (CXCL12) produced by stromal cells [156], molecular inhibition of Rac3 prolonged the survival of mice with BCR/ABL induced leukaemia [116]. Similarly, central nervous system metastatic pre-B leukaemia cells (SD1-cells) overexpress Rac2, and when pre-treated with the Rac inhibitor, NSC23766, prior to engraftment, delayed disease establishment and significant reduction in leukaemia burden in extramedullary organs and the cranium [154]. The tetraspanin family member, CD9, is also reported to activate Rac1 and form cytoplasmic extensions and homing of B-ALL cells in vivo and in vitro [157]. Although the direct role of Rac proteins in ROS production has not been discussed, their reported role in NOX1-3 activation [117,158] raises the prospect that they may contribute to NOX mediated ROS production in ALL; an intriguing possibility that warrants further exploration.

### 5.9. NAD(P)H Quinone Dehydrogenase 1 (NQO1)

The NAD(P)H:quinone oxidoreductase 1 (NQO1) is a cytosolic two-electron oxidoreductase, detoxification and cytoprotective enzyme that protects cells against oxidative stress [159,160]. This enzyme functions as a superoxide reductase to generate antioxidant forms of ubiquinone and vitamin E to protect cells against oxidative stress. A polymorphism (C609T) in the *NQO1* gene leads to loss of protein expression, and hence loss of downstream ROS detoxification, with heterozygous (C/T) and homozygous (T/T) alleles reducing and abolishing reductase activity, respectively [161,162,163]. Studies report the increased association of low/null NQO1 activity and increased risk of infant ALL [162,164]. The loss/reduction in reductase activity is potentially associated with in utero genotoxic stress elicited through various oxidative cascades that accompany foetal haematopoiesis in the yolk sac from where it migrates to form part of the foetal bone marrow [165]. Conversely, overexpression of NQO1 is cytoprotective, and known to drive metastasis [166], with overexpressed NQO1 commonly detected in late stage and poor prognoses cancers [167]. Depletion of *NQO1* sensitised treatment-recalcitrant non–small cell lung cancers to low-dose radiotherapy by decreasing the cells reductase activity and increasing the genotoxicity of ionising radiation [168]. Furthermore, knockdown of *NQO1* in immune checkpoint inhibitor resistant tumours increased innate immune response to stimulate antitumour T cell adaptive immunity, and when retreated with checkpoint blockade therapies, eradicated therapy-resistant tumours [27].

## 6. Redox Homeostasis in ALL

ALL cells have evolved several compensatory mechanisms to ensure that ROS production does not induce irreversible DNA damage and cell death [169] (Figure 1). Many of these mechanisms centre on the expression of antioxidant systems with, for example, the gene and protein levels of thioredoxin reductase 1 (TXNRD1), TXN1 and PRDX1 all being elevated in B-ALL cell lines and primary patient samples [170]. Similarly, although H_2_O_2_ is usually a key signal in dexamethasone-induced apoptosis, pre-B-ALL cells that display dexamethasone resistance are characterised by overexpression of GSH; yet can be re-sensitised using L-buthionine-(S, R)-sulfoximine (BSO), an inhibitor of GSH synthesis [171]. Furthermore, thymic lymphoma cells that overexpress catalase are also resistant to dexamethasone [172]. As previously noted, manganese superoxide dismutase (MnSOD) overexpression drives H_2_O_2_ production and thus sensitises thymic lymphoma cells to dexamethasone via the release of mitochondrial cytochrome c and activation of caspases [173]. Importantly, increased GSH expression is associated with the increased risk of relapse in children diagnosed with ALL [174]. High-level expression of *GPx1* is also seen in ALL, influenced by decreased expression of *miR-491-5p* and *miR-214-3p* via VPS9D1 antisense RNA 1 [175]. Knockdown of *GPx1* reduced proliferation and activated apoptosis, with the *VPS9D1* antisense RNA 1 acting as a tumour promoter to increase *GPx1* expression and decrease *miR-491-5p* and *miR-214-3p*.

Regardless of the subtype, nuclear factor-κB (NF-κB) complexes show constitutive activation in paediatric ALL [176,177]. NF-κB proteins are a family of transcription factors that regulate immune responses to pathogens, inflammation, promote growth and proliferation, and cell development [178]. Indeed, NF-κB transcription factors are responsible for the expression of the ROS producing NADPH oxidase enzymes [179] (discussed in Section 3, Section 5.2, Section 5.7, Section 5.8 and Section 7.2) but can also upregulate the expression of antioxidants (reviewed in [60]). Not only does NF-κB activity increase NADPH oxidase expression and hence ROS production, but ROS regulates the transcriptional activity of NF-κB through degradation of the NF-κB inhibitory protein IκB, promoting nuclear translocation and transcription of κB genes, creating both a feed-forward and positive feedback loop [180,181]. Potentially, the chronic activity of PI3K-Akt signalling driven by recurring ALL associated somatic mutations (BCR/ABL, *IL7R*/*CRLF2*), may help to initiate the NF-κB positive feedback loop. Akt activates IκBα kinase (IKK) and p38 MAPK leading to the phosphorylation and degradation of IκB [182]. Loss of PTEN expression further potentiates NF-kB activity through unfettered PI3k/Akt signalling driving the activity of IKK, and further degradation of IκB, supressing apoptosis [183] and driving resistance to doxorubicin [184].

The observations that NF-κB acts an oncogene in ALL, contrasts with pre-B-ALL studies suggesting NF-κB1 is a tumour suppressor. The ratio of phosphorylated and hence nuclear translocated STAT5 to RELA expression (NF-κB transcriptional effector) correlates with B-ALL patient survival and disease remission [185]. Competition between STAT5 and NF-κB for common binding sites increased expression of STAT5 genes including Cyclin D2 and D3 (*CCND2*, *CCND3*) and the oncogene MYC (*MYC*) [185], driving an aggressive form of B-ALL. As there is no doubt that STAT5 plays a critical role in the leukaemogenesis of B and T -ALL, with its activity required for transformation downstream of ALL oncogenes [186,187,188], it is interesting to postulate the direct roles ROS plays in the activity of STAT5 and vice versa. Like NF-κB, STAT5 enhances transcription of the NOX (specifically NOX4). By doing so, STAT5 promotes increased ROS production which acts as a feed-forward loop. However, it is unknown whether the activity of STAT5 plays a direct role in NOX complex activation in ALL. In AML, phosphorylated STAT5 has also been shown to co-localise with Rac1, suggesting a mechanism in which phosphorylated STAT5 promotes ROS production by NOX. Given Rac1 is overexpressed in primary ALL and AML primary blasts compared to controls [189], and pharmacological inhibition of Rac1 is selectively cytotoxic to primary ALL cells and not on normal lymphocytic cells [190], it is indeed possible that the high-level activity of STAT5 in ALL may promote Rac1- induced NOX activity helping to form a feed-forward loop analogous to AML.

There is clear evidence of oxidative dysfunction in ALL. However, to ensure ROS accumulation does not exceed the tipping point and shift from their leukaemogenic roles to induction of regulated cell death pathways, these malignant cells hijack homeostatic mechanisms for survival. The Nuclear factor-erythroid factor 2-related factor 2- NRF2 (*NFE2L2*), a transcription factor negatively regulated by Kelch-like ECH-associated protein 1 (*KEAP1*) under basal conditions. ROS mediated redox cysteine PTMs induce conformational changes in KEAP1, leading to the release and subsequent translocation of NRF2 to the nucleus. Within the nucleus, NRF2 binds to antioxidant response element loci located at the promoters of multiple genes that orchestrate cytoprotection through rapid expression of *NQO1* (discussed in Section 5.9), combatting oxidative stress and driving cell survival [191,192]. A recent study showed 73% of paediatric ALL patients (22/30) harboured nucleotide changes in genes mapping to the KEAP1/NRF2/NF-κB1/p62 pathway [193]. The significant functional crosstalk between NF-κB and NRF2 suggests that both play important roles in the oxidative dysfunction of ALL cells. It is interesting to note that ALL cells are afforded protection against standard induction chemotherapies via interaction with neighbouring adipocytes [194]. In this regard, daunorubicin treatment of ALL cells has been shown to dramatically upregulate NRF2-mediated oxidative stress response in adipocyte co-cultures and protect ALL cells from genotoxic stress. Such data implies that ALL cells induce oxidative stress in adipocytes through yet an unknown mechanism. Blocking GSH synthesis in adipocytes subsequently re-sensitises ALL cells to daunorubicin, suggesting adipocyte secreted exogenous antioxidants protect ALL cells from chemotherapy [195]. In a similar context, mesenchymal stem cells (MSCs) found in bone marrow release thiols (antioxidant) to protect T-ALL cells from parthenolide induced oxidative stress [196].

In addition to intracellular factors, ALL cells have been reported to make direct contact with the bone marrow stromal cells via tunnelling nanotubes (TNTs); long cylindrical non-adherent actin-based cytoplasmic extensions that play an important role in direct communication and transfer of macromolecules between adjacent cells [197]. Recently, Jurkat ALL cells were reported to directly transfer mitochondria to the bone marrow stromal cells via TNTs upon exposure to chemotherapeutic drugs, thereby reducing ROS induced cellular death [198]. Furthermore, primary patient derived pre-B-ALL cells signal to the bone marrow stromal cells through TNTs, driving secretion of pro-survival cytokines such as interferon-γ–inducible protein 10/CXC chemokine ligand 10 (*CXCL10*), IL-8, and monocyte chemotactic protein-1/CC chemokine ligand (*CCL2*) causing resistance to prednisolone [199].

T-ALL switch their metabolic programs in a similar way to normal HSCs when cultured in low oxygen [200]. Reduced mitochondrial activity and cell cycle progression in these low oxygen niches increases glycolysis and lowers their sensitivity to vincristine and cytarabine (cell cycle-related drugs) and dexamethasone, compared with T-ALL cells grown under normoxic conditions [201]. While low oxygen levels suppressed the activity of mTORC1, it increased the activity of HIF1α with the concomitant increase in the expression of HIF1α effector genes such as, *VEGF*, *GLUT3* and *CXCR4* to reduce mitochondrial activity and as such ROS levels in ALL [201].

In contrast, B-ALL cells seem to rely more on oxidative phosphorylation and mitochondrial activity than T-ALL [34]. B-ALL cells with reduced NADP/NAD^+^ ratios were enriched for functional leukaemia-initiating cells (LICs) resistant to cytosine arabinoside (Ara-C) [34]. These cells maintained their oxidative stress levels and resistance to Ara-C through the activation of pyruvate dehydrogenase complex component X (*PHDX*). Further, Ara-C resistance was attenuated by suppressing oxidative phosphorylation using venetoclax, metformin, and berberine, inhibitors of mitochondrial metabolism in vitro and in vivo [34].

Autophagy is another mechanism that aids leukaemic cells to survive oxidative stress-induced apoptosis. By way of example, ROS have been shown to induce the expression of Beclin-1 (*BECN1*) (an autophagy related protein with an essential role in autophagosome formation) and increase the removal of injured mitochondria to drive chemotherapy resistance in ALL [202,203]. Importantly, quinacrine (QC) (an anti-malaria drug that potently inhibits autophagy) in combination with vorinostat (a pan-histone deacetylase (HDAC) inhibitor), significantly increased ROS production, which reduced autophagy and caused synergistic apoptosis in T-ALL cells [204].

## 7. The Modulation of ROS to Target ALL

### 7.1. Pro-Oxidant Therapies

The importance of ROS in driving the leukaemogenesis of ALL highlights the potential for modulating the redox balance of malignant B or T lymphoblasts to improve treatment. Given their characteristically high-ROS chemotype and dysregulated antioxidant system, further redox modulation is an effective strategy to eliminate ALL blasts (Figure 2) [205]. Accordingly, many therapeutics used in the treatment of ALL increase oxidative stress. For example, the antimetabolite methotrexate, microtubule inhibitor vincristine, and DNA damaging agents such as doxorubicin, daunorubicin and cytarabine drive oxidative stress [206,207,208,209]. However, to enhance the therapeutic benefit of these strategies, simultaneous modulation of the antioxidant system such as NRF2 (discussed in Section 6) may be necessary to decrease resistance and reduce clonal evolution.

As mentioned above, dexamethasone works by binding to the glucocorticoid receptor driving ROS production via the endoplasmic stress pathway and ultimately causing DNA DSBs and error-prone repair [210,211]. Multiple other targeted therapies have been implicated in modulating the redox system to induce apoptosis including the Bcl-2 inhibitor venetoclax [212], HDAC inhibitor vorinostat [213], antimalarial drug quinacrine (QC) [204], antidiabetic drug metformin [214], Ca^+^ and K^+^ channel activator NS1619 (benzimidazolone) [215], TXNRD1/2 inhibitor auranofin (AUR) [216], PRDX1/2 and TXN1 inhibitor adenanthin (ADE) [170] (Table 1 and Figure 2).
antioxidants-10-01616-t001_Table 1Table 1Pro-oxidant therapies used in the pre-clinical models of various cancers.Drug(Status)SignallingPathwaysPossible Mechanism of ActionReferencesArsenic trioxide (FDA approved drug for AML)Mitochondrial membraneNOX2↑ ROS by inhibiting reducing GSH expression, and inhibiting GPx, GST and catalase activity, and increasing NOX2 expression and activity[217,218]DoxorubicinMitochondria↑ ROS due to topoisomerase-IIβ transcriptome mediated loss of ΔΨm[219]Tigecycline(FDA approved drug for disease other than cancer other disease)Mitochondria↑ ROS by suppressing the translation of complex I and IV of mitochondrial proteins[220]NOV-002(In Clinical trials for Myelodysplastic Syndrome (MDS) and NSLC)Redox enzymes↑ ROS by acting as GSSG mimeticAlso stimulates blood cell production[221]2-methoxyestradiol (Panzem)(FDA approved drug for multiple cancers including meyloma)Redox enzymes↑ ROS by inhibiting Superoxide dismutases (SODs)[222]ATN-224(Completed phase 2 clinical trials for MM and prostate cancer)Redox enzymes↑ ROS by inhibiting copper/zinc SOD activity[223]Imexon(Completed phase 2 clinical trials for multiple myeloma and lymphoma)Redox homeostasis↑ ROS by reducing cysteine and GSH pool[224]PX-12(Completed phase 2 clinical trials to treat solid tumours)Redox enzymes↑ ROS by inhibiting thioredoxin enzymes system[225,226]Parthenolide and derivatives(Natural compound)(Phase I/II studies for various cancers)Thiol inhibitors↑ ROS by depleting cellular thiol including GSH[227]BSO (L-buthionine-(S,R)-sulfoximine (BSO)Redox system↑ ROS by depleting GSH[228] TPEN (zinc chelator)Redox System↑ ROS and induces loss of ΔΨm by unknow mechanism[229,230]Cannabidiod CP55940(Natural compound)Redox system↑ ROS loss ΔΨm and Ca^2+^ overload[231]Curcumin (Natural compound)Mitochondria↑ ROS loss of ΔΨm by unknow mechanism[232]VinblastineRedox system↑ ROS by depleting GSH levels[233]Quercetin(Natural compound)Mitochondria↑ ROS by depleting GSH levels and inducing loss ΔΨm [234]Erastin (eradicator of RAS and ST)Redox system↑ ROS by inhibiting and Cys2/glutamate antiporter leading glutathione depletion [235]Auranofin(FDA approved drug to treat rheumatoid arthritis)Redox enzyme↑ ROS by inhibiting TXNRD[236]Adenanthin (ADE)(Natural compound)Redox enzymes↑ ROS by inhibiting PRDX1/2 activities [237]RSL3 (RAS-synthetic-lethality 3)Redox enzymes↑ ROS by inhibiting GPX4 and induce ferroptosis[238]Matrine(Natural compound)Mitochondria↑ ROS, loss ΔΨm and mitochondrial swelling by unknow mechanism[239]NS1619Mitochondria↑ ROS production by unknow mechanism[215]AdaphostinMitochondria↑ ROS and loss of ΔΨm by inhibiting mitochondrial respiration[240]APR-246(Phase I clinical trials)Redox enzymes↑ ROS by inhibiting TRX1 and glutaredoxin[241]Sanguinarine(Natural compound)Mitochondria↑ ROS and loss of ΔΨm by depleting glutathione [242]Alantolactone(Natural compound)
↑ ROS by inhibiting glutathionereductase (GR)[243]CB-839Redox enzyme↑ ROS by inhibiting glutaminase to suppress glutathione production[244]6-Shogaol(Natural compound)Unknown mechanism↑ ROS by reducing GSH[245,246]Note: the list is not exhaustive. ΔΨM, mitochondrial membrane potential; TRX, thioredoxin; GRX, TXNRD; thioredoxin reductase; ↑, increase.


A by-product of mitochondrial oxidative phosphorylation is ROS production [247]. Therefore, mitochondrial electron transport chain complexes represent logical targets for leukaemia specific therapies. One such class of drugs are the mitocans that target the mitochondria of cancer cells [248], and include tigecycline an FDA-approved agent that has shown promising anti-leukaemic results through decreased fidelity of the mitochondrial respiratory complexes I and IV, driving oxidative stress, DNA damage, and apoptosis [220,249] (Figure 2). Alternatively, the imipridones ONC201, ONC206 and ONC212 bind with, and activate mitochondrial caseinolytic protease P ClpP (*CLPP*), a protease that degrades misfolded proteins within the mitochondria. Activation of ClpP has been shown to drive selective degradation of respiratory chain protein substrates and disrupt membrane potential (ΔΨm) resulting in increased mitochondrial ROS and apoptosis in Z-138 lymphoma cells [250]. Our own unpublished data shows ONC201 is highly effective in inducing apoptosis in primary ALL blasts. Building on these observations, analogues of vitamin E such as α-tocopheryl succinate (α-TOS), can inhibit the succinate dehydrogenase (SDH) activity of complex II (CII), thereby inducing ROS production and driving apoptotic cascades [251]. Similarly, D-α-tocopheryl polyethylene glycol 1000 succinate (TPGS), has been reported to induce oxidative stress by modulating mitochondrial membrane potential resulting in apoptosis [252]. Some mitocans are in clinical trials, while others already have FDA approval [248].

Various natural compounds have also been reported to increase ROS production in cancer cells. Resveratrol (a phenol found in relatively high concentration in berries) and curcumin (obtained from turmeric) are two well-known examples of such compounds. Resveratrol has been reported to reduce ΔΨm, driving caspase activity and cell death in ALL [253,254]. Similarly, curcumin elevates intracellular ROS through binding and possibly modulating the activities of antioxidant enzymes in leukaemia [255]. Curcumin-induced apoptosis in a panel of ALL cell lines via inhibition of PI3K/Akt signalling increased ROS production and release of pro-apoptotic cytochrome c protein. Further, suboptimal doses of curcumin has been shown to enhance the anticancer activity of cisplatin [232]. A range of other natural anticancer compounds, together with their mode of action, are listed in Table 1.

### 7.2. Therapies to Reduce ROS

Pro-oxidant therapies are effective anti-ALL treatments, however, work by increasing global oxidative stress and hence tend to be very non-specific. Given the dynamic redox-balance of HSCs; it is challenging to employ a pro-oxidative treatment approach in leukaemia without causing systemic toxicities [147].

An alternative strategy is to reduce the activity of oxidase enzymes responsible for ROS production, and or target the signalling pathways that drive the activity/expression of these oxidases (Figure 3). For example, azelaic acid (ZA) is a natural compound that inhibits tyrosinase, cytochrome-P450 reductase and respiratory chain enzymes, thereby suppressing the production and actions of ROS. Illustrative of this, ZA increased the expression of PRDX2/PRDX3 antioxidant enzymes, driving cell cycle arrest at G1 and apoptosis in AML cells in vitro and in vivo [256]. Similarly, metformin, has been shown to induce cytotoxicity in cancer cells by decreasing ROS levels and suppressing mitochondrial ATP synthesis; a response attributed to metformin’s inhibition of mitochondrial respiratory complex I (CI) [214]. Nevertheless, to date no clinical trials have employed antioxidant therapies alone or in combination to specifically treat ALL.

In contrast to ROS produced as a by-product of metabolism, NOX enzymes produce ROS as a primary function in response to infection [25] or to drive angiogenesis following hypoxia [257]. Further, NOX has been implicated in tumour progression [134,258,259,260,261], leukaemic stem cell (LSC) self-renewal [32] and resistance to therapies [262]. Cell lines harbouring FLT3 internal tandem-duplication (FLT3-ITD) mutations, showed increased expression and activity of NOX2 linked with elevated ROS-associated DNA damage. Partial knockdown of the transcription factor STAT5 (*STAT5A* and *STAT5B*) in FLT3-ITD cells resulted in decreased ROS production [263], highlighting the oncogenic association between recurring somatic mutations and the necessity for leukaemia cells to maintain a state of oxidative dysfunction.

Due to the emerging role of NOX in various cancers, targeting of NOX alone or in combination with the other therapies is gaining popularity as a treatment paradigm [264,265,266]. Genetic or pharmacological (GKT137831) inhibition of NOX4 reduced cancer associated fibroblasts growth and survival [243] and metastasis of non–small cell lung cancer cells in vivo and in vitro [266], and potentiated sensitivity to immunotherapy in murine lung cancer models [265]. Similarly, genetic/chemical inhibition of NOX suppressed the growth of gastric [261], colon [267], skin [268], liver [269] and human granulosa-lutein and granulosa [270] tumours.

Given the increased expression and/or activity of NOX (discussed in Section 3), targeting NOX is a potential therapeutic approach particularly for ALL blasts residing within the hypoxic bone marrow niche [114,115,134]. Importantly, NOX2 derived ROS has been reported to promote transfer of mitochondria from the bone marrow stromal cells to the leukaemic cells via TNTs [271]. Therefore, combining NOX inhibition with standard of care chemotherapies may be an effective strategy to target blasts and LICs/LSCs protected by the bone marrow, to overcome the potential of drug resistance [271] (Figure 3). Multiple compounds have been reported to inhibit NOX enzymes and promisingly, some of these compounds are already in clinical trials to treat various clinical conditions [272,273,274] (Table 2). For example, NOX proteins drive oxidative stress and multiple organ damage exacerbating diabetic complications such as diabetic retinopathy, nephropathy, neuropathy, disorders of the cardiovascular system [275] and acute pancreatitis [276]. It follows that NOX inhibitors such as GKT137831 (NOX1 and NOX4 inhibitor), APX-115 (pan NOX and DUOX inhibitor) and GSK2795039 (NOX2) have shown promising results in preclinical mouse models of disease [274,277,278,279,280], with some currently in clinical trials (Table 2).

Despite promising results in preclinical tumour models, including ALL, no clinical trials are yet evaluating the efficacy of NOX inhibition to treat cancer either alone or in combination. In taking up this challenge, unpublished work from our own group shows the potential of NOX inhibitors (GSK2795039 and APX-115) used in combination with tyrosine kinase inhibitors to target ‘kinase-active’ ALL cells in vitro and in vivo (unpublished data). Given that at least some NOX inhibitors are already in clinical trials for other indications, studies that provide much needed safety and/or tolerability data, targeting NOX is the natural next step in the development of new ALL treatment regimens.

## 8. Conclusions

There is now a compelling body of evidence that ROS play an important secondary messenger role in the modulation of various oncogenic signalling pathways associated with the initiation and progression of different cancers, including ALL (Figure 1). There is also conclusive evidence that redox dysfunction plays an important role in the leukaemogenesis of ALL. Metabolic adaptation, mitochondrial transfer, hypoxia, activation/upregulation/constitutive activation of oncogenic signalling pathways and dysregulation of cellular antioxidant systems all converge to drive a state of pro-oxidation, which propagates further insult to the genome and leads to clonal evolution, treatment resistance and poor outcomes. Capitalising on this knowledge, pro-oxidant therapies have had success in the treatment of ALL, however, this also leads to significant toxicity in healthy cells with potential short- and long-term complications. Therefore, an improved understanding of the sources and function of ROS in ALL is necessary to identify therapeutic modalities that selectively regulate the dynamic redox state of ALL to improve treatment and outcomes. In this context, a promising line of investigation is the application of antioxidant therapies, specifically those targeting the NOX family of enzymes, which represent an exciting drug target in ALL that has yet to be extensively explored.

## Figures and Tables

**Figure 1 antioxidants-10-01616-f001:**
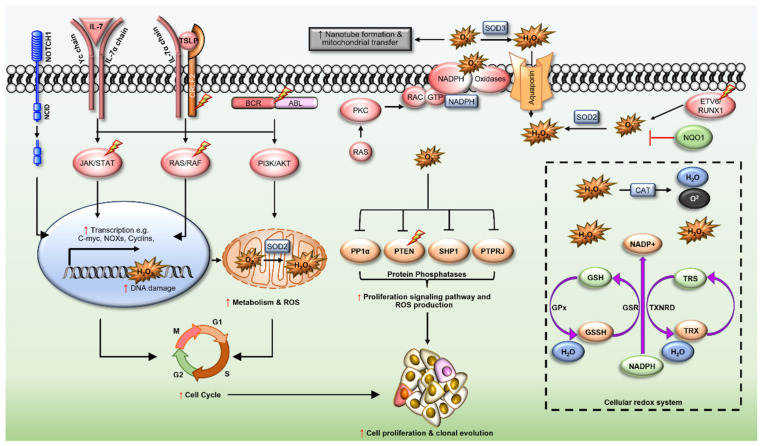
Signalling pathways linked to reactive oxygen species production in acute lymphoblastic leukaemia. Recurring somatic mutations to *CRLF2*, *JAK*, *NQO1*, *NOTCH1* and *RAS*, chromosomal translocations such as BCR/ABL and ETV6/RUNX1 as well as overexpression of CRLF2 and IL7 receptor drive excessive production of intracellular reactive oxygen species (ROS) production (superoxide- O_2_^.−^ and hydrogen peroxide- H_2_O_2_) in acute lymphoblastic leukaemia (ALL). High-level ROS production drives redox signalling through oxidative posttranslational modifications that increase the activity of kinases and inactivate protein tyrosine phosphatases, and cause lipid peroxidation and genomic instability leading to leukaemia progression and chemotherapy resistance. Cellular redox systems (shown in the hashed rectangle) regulate ROS homeostasis by converting H_2_O_2_ to water and suppress ROS induced apoptosis. Red shapes = proteins with increased activity or expression; dark green = proteins with decrease activity or expression; orange = oxidised proteins; light green = reduced proteins.

**Figure 2 antioxidants-10-01616-f002:**
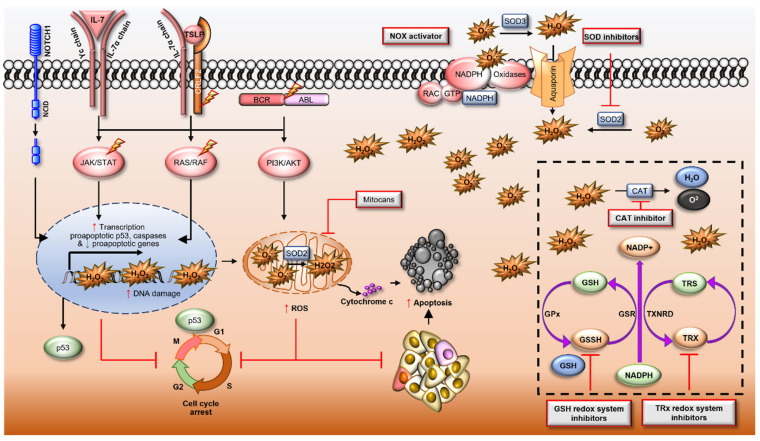
Pro-oxidant therapies for the treatment of acute lymphoblastic leukaemia. Pro-oxidant therapies such as chemotherapies cause redox dysfunction driving irreversible genotoxic stress and are linked to increased activity of NOX proteins. Mitocans target the mitochondrial electron transport chain, increase the permeability of mitochondrial membranes, increasing cytoplasmic ROS. High-level ROS production increases expression and activity of pro-apoptotic proteins such as p53 and caspases, inducing cell cycle arrest, releasing cytochrome c and inducing apoptosis in leukaemic cells. The molecular targets and mechanisms of drugs enhancing ROS production are listed in Table 1. Red shapes = proteins with increased activity or expression; lightning bolt = proteins harbouring mutations/translocations; orange = oxidised proteins; light green = reduced proteins.

**Figure 3 antioxidants-10-01616-f003:**
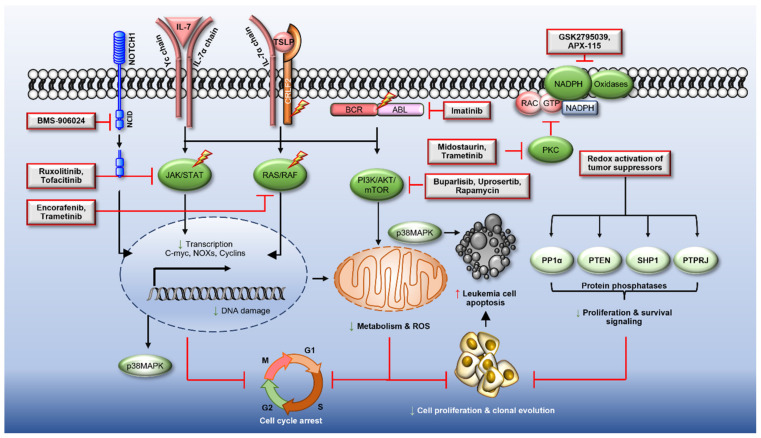
Therapeutic strategies targeting reactive oxygen species production for the treatment of acute lymphoblastic leukaemia. Therapies targeting oncogenic kinases in acute lymphoblastic leukaemia (ALL) reduce ROS production. NOX inhibitors (NOXi) used in combination with targeted therapies synergise with ALL tyrosine kinase inhibitors (TKIs) or kinase inhibitors (KIs) and are emerging as a novel therapeutic strategy. Red shapes = proteins with increased activity or expression; dark green shapes = decreased activity or expression; lightning bolt = proteins harbouring mutations/translocations; orange = oxidised proteins; light green = reduced proteins.

**Table 2 antioxidants-10-01616-t002:** NOX inhibitors.

Drug	Possible Mechanism of Action	Impact on NOX	Experimental Model	References
Normobaric oxygen (NBO) and Hyperbaric oxygen (HBO)	Unknown	NOX2 levels and activity downregulation	Male Sprague–Dawley rats	[281,282]
Ethanol	Unknown	Reduced activity enzyme and gp91 expression	Rat model of ischemia	[283]
PKC inhibitors (Calphostin C, Chelerythrine and Ruboxistaurin mesylate)	Inhibiting PKC mediated p47*^phox^* phosphorylation	Reduced activity of NOX	Human polymorphonuclear leukocytes	[284]
Diphenylene iodonium (DPI)	Forms redox adduct with the NOX catalytic core	Reduced NOX 1 expression	Human colon carcinoma cells in vivo and in vitro	[285]
Apocynin (4-hydroxy-3methoxy-acetophenone), natural compound)	Blocks p47*^phox^* cell membrane migration and NOX assembly	Inhibit assembly and activity of NOX1 and NOX2	Human neutrophils	[286]
VAS2870 (Vasopharm),	Upregulation of NOX2 and 4 targeting microRNA	Reduced expression of NOX2 and Nox 4	Rat model of ischemia	[287]
Celastrol	Disrupt NOX enzyme assembly	Binds to p47*^phox^* and disrupted p22*^phox^*	Human neutrophils, HEK293 and CHO in vitro and cell free assays	[288]
NOX peptide inhibitors (e.g gp91ds-tat)	Disrupt NOX enzyme assembly	Bind with NOX subunits disrupting NOX assembly	Cell free and cell-based assays	[289]
GSK2795039	Compete for NADPH binding site	Specifically inhibit NOX2 by excluding NADPH binding	Cell based, cell-free assays and animal model of acute pancreatitis	[277]
Ebselen and analogue	Blocks p47*^phox^* cell membrane migration and NOX assembly	Inhibits NOX1 and NOX2 assembly	Human neutrophil and cell-free assays	[290]
GKT137831 (Phase 2 clinical trials for diabetic complications)	Direct interaction with NOX complex	Inhibits NOX1/4 activity	Cell-free assays and animal models	[268,291]
APX115 (Phase 1 clinical trials) (Phase 2 clinical trials for Covid-19)	Unknown	Decreases the expression of NOX1-3 proteins	Diabetic mouse model	[278]
Aspirin (FDA Approved drug)	Unknown	Lowers Nox 4 enzyme	Human endothelial cells	[292]

Note: the list if not exhaustive.

## Data Availability

Data is contained within the article.

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
