# Peer review of "Reactive Oxygen Species in Acute Lymphoblastic Leukaemia: Reducing Radicals to Refine Responses"

_antioxidants, 2021, doi:10.3390/antiox10101616_

Round 1

Reviewer 1 Report

This is an outstanding. well written, thorough and expansive review of the role of Reactive Oxygen Species, their role in hematopoiesis and the ways that they are dysregulated in various subsets of acute lymphoblastic leukemia associated with specific molecular lesions. There is also a fascinating summary of the various types of ROS modulating therapy and how these apply to cancer therapy and ALL therapy in particular. Figures and Tables are well made and thorough and referencing is accurate throughout.

Author Response

On behalf of the authors, we sincerely thank Reviewer 1 for their encouraging comments.

Reviewer 2 Report

The authors reviewed the molecular mechanisms underpinning overproduction of ROS in ALL, and their roles in disease progression and drug resistance. They also discussed strategies to target ROS production, with a specific focus on the NOX enzymes, to improve the treatment of ALL.

The authors should summarize the specific ROS-sensing signaling pathways that mediate the ALL-regulated cellular functions. Is there any differences in comparison with the acute myeloid leukemia?

In the in some paragraphs it is not clear who drives the ROS and which is the ROS-activated  signaling pathways. It is well known that leukemia cells exhibit elevated ROS levels, of which several identified pathways further increase ROS generation. The authors should analyze further this point.

What about the ROS overproduction and stimulated innate and adaptive immunity? 

The pro-oxidant and antioxidant approach is not well described

Author Response

We are sincerely grateful for the expert comments of Reviewer 2 (R2). In our revised manuscript we have addressed each of these comments, which has increased both the scope and depth of our review. Below I summarise the information that has either been added and/or refined.

1. R2: Specific ROS-sensing signalling pathways that mediate ALL cellular functions. Response: As discussed in our article, one of the major redox sensing pathways is that regulated by the Nuclear factor-erythroid factor 2-related factor 2- NRF2 (NFE2L2) transcription factor, which drives expression of the antioxidant response elements, leading to cytoprotection even under situations of very high-level oxidative stress. What is less covered in our original review is the importance of the Nuclear factor-κB (NF-κB) transcriptional complexes, which show constitutive activation in paediatric ALL [1]. This is particularly important due to the significant crosstalk between NF-KB activity and NRF2. Further, NF-κB forms a positive feedback loop with NADPH oxidases enzymes to produce ROS [2]. We have added this important additional data to further highlight how ALL oncopathways may play a direct role in both ROS-sensing and ROS production.

2. R2: Additionally, the major oncogenic ROS sensing and stimulating pathways in ALL include PI3K/Akt and JAK/STAT. Response: We have expanded these sections to discuss the way these oncogenes create a ROS feed forward and positive feedback loop, and draw the reviewer’s attention to lines: 204-210, 215-227, 243-249, 309-316, 323-331, 345-359, 371-375, 381-383, 389-397, 492.

3. R2: It is not clear who drives the ROS, and which is the ROS-activated signaling pathways. Further, it is well known that leukemia cells exhibit elevated ROS levels, of which several identified pathways further increase ROS generation. Response: We thank the reviewer for highlighting the complex interplay between ROS production, sensing and oncogenic signalling. In many cases, the original research articles provide descriptive rather than functional observations linking high-level ROS production and the commensurate ALL associated oncogenic signalling. To provide more clarity, we have added additional information regarding the pathways driving ROS production, e.g., STAT5 (lines 246-252, 313-317, 494-512, 662-665), NF-kB (lines 279-282, 484-497, 520-527, loss of PU.1 (377-389).

4. R2: What about the ROS overproduction and stimulated innate and adaptive immunity? Response: This is a wonderful suggestion and given the breadth of research in this field, this topic could form the focus of a stand-alone review. To address the Reviewer’s comment within we have taken the opportunity to expand section 4: The essential role of ROS in the maintenance of haemopoietic stem cells (HSCs), and innate and adaptive immunity, in which we now provide an overview of the roles that immune cells play and in ROS overproduction and B and T cell stimulation (lines 262-292).

5. R2: The pro-oxidant and antioxidant approach is not well described. Response: We sincerely apologise for the lack of clarity in this section. To address this concern, we have reduced the complexity and wordiness of this section. Please see lines 573-580, 593, 595, 603-613, 617-620, 625-630, 636-639, 642, 646-647, 670, 674, 678, 681-695.

Reviewer 3 Report

The manuscript of Mannan and coworkers was aimed at reviewing the role of ROS in ALL and their possible role in improving the disease therapy. Although of interest and overall well written, the subject matter is lacking of novelty since a quite similar review has been already published in May 2021 (Chen et al., “Redox Control in Acute Lymphoblastic Leukemia: From Physiology to Pathology and Therapeutic Opportunities”. Cells (mdpi journal) 2021, 10, 1218. https://doi.org/10.3390/ cells100512)

Author Response

We would like to thank Reviewer 3 for the kind comments regarding the quality and interest of the manuscript. However, we are disappointed that we have not made the novelty of our manuscript clearer.

The review published early this year “Redox Control in Acute Lymphoblastic Leukemia: From Physiology to Pathology and Therapeutic Opportunities” provides an excellent summary of the roles ROS plays in HSC activity and in this regard has some similarity to our review of AML published in 2019 (3390/ijms20236003). To provide the necessary background on the roles ROS plays in the homeostasis of the cells of the haematopoietic system it is indeed necessary to revisit the function of ROS in immune cell development, hence there are some unavoidable overlaps. However, the stated intention of “Redox Control in Acute Lymphoblastic Leukemia: From Physiology to Pathology and Therapeutic Opportunities (referred to as 10.3390/cells10051218 from hereon) is to exploit the redox environment of ALL cells using pro-oxidant therapies and accordingly, this article pays particular attention to antioxidant mechanisms that help leukaemia cells escape damage caused by pro-oxidant therapies. This coverage includes an in-depth summary of the regulatory mechanisms of redox homeostasis in normal and malignant HSCs to aid in the development of more effective treatment plans. This review also provides an excellent summary of the bone marrow microenvironment, which we did not discuss in our article. Additional points of difference include:

    1. A strong point of difference is our focus on therapies that suppress ROS production, especially those targeting ROS producing enzymes such as the NADPH oxidases (NOXs), in recognition of their emergence as promising targets to treat cancers. This is a major focus of the research of my team and a treatment approach we believe can be exploited to improve the treatment of ALL. To help clarify the difference in the reviews from the outset, we have amended our title to include ‘Reducing Radicals to Refine Response’ and included additional information within the summary sentence of our Introduction to highlight the focus of our article.
    2. Our review is focused on the biochemical effects of elevated ROS linked to the specific and recurring genomic profile of ALL, and how ROS regulates signalling proteins that contribute to the ALL phenotype.
    3. Our Introduction provides an overview of ALL, prognosis, treatment option and risk factors and genetic abnormalities; noting that, with the exception of general and very broad points 3390/cells10051218 does not introduce ALL to the readers. We believe this is important background information and accordingly, discuss in detail each of the recurring genomic events and their association with redox dysfunction (Sections 1 and 5).
    4. Importantly, our review provides an comprehensive overview of the chemical nature of ROS and includes examples of ROS impacts signalling in multiple diseases including ALL. 3390/cells10051218 did not touch on the biochemistry of reactive molecules.
    5. 3390/cells10051218 provides a limited description of NOX and the respiratory burst of myeloid cells, but does not consider tissue specific expression, different isoforms and subunits of NOX nor the role of aquaporins, each of which have been covered in detail in our review. Further, in answer to the excellent suggestions of Reviewer 2, we provide further information on the roles of ROS in adaptive and innate immunity to highlight how ROS regulates B and T cell function.
    6. Additionally, suggestions from Reviewer 2 have helped to define the proteins and pathways that act as ROS sensors and ROS drivers, which provides mechanistic information about autocratic redox dysfunction in ALL.
    7. Except for some limited overlaps in general signalling and associated enzymes, our description of ROS influenced redox signalling is very different. For example, 3390/cells10051218 does not discuss Cysteinyl residues, NADPH and glucose metabolism in the redox system, which we believe are implicit in the oncogenic function of ROS in ALL. The roles of PTEN/PI3K/Akt in the direct activation of NOX and how constitutive activation of PI3K/Akt downstream of numerous ALL associated oncogenes directly influences NOX protein expression to create not only a positive feedback loop but also feed-forward mechanism.
    8. 3390/cells10051218 does not discuss important ROS associated signalling protein “Cytokine receptor-like factor 2 (CRLF2)” particularly important as patient diagnosed with Ph-like ALL commonly harbour alterations to the signalling nexus and most commonly show increase rates of treatment failure, highlighting the potential of targeting NOX in combination with standard therapies. Further, 10.3390/cells10051218 does not discuss IL7R mutations and the associated hyperactivation of downstream pathways. The role of ROS induced PTM to positively regulate PI3K/Akt pathways.
    9. 3390/cells10051218 does not discuss the RAS pathway, which is a known driver of ROS production and poor ALL outcomes.
    10. 3390/cells10051218 is missing the important antioxidative enzyme “NAD(P)H Quinone Dehydrogenase 1 (NQO1)”, the loss of which is associated with ALL, further driving the activity of redox sensitive oncogenes.
    11. 3390/cells10051218 does not discuss TXNRD1, TXN1 and PRDX1 expression in ALL. It also does not give consideration to how elevated ROS can activate NRF2, autophagy as strategy to overcome ROS induced apoptosis, possible differences between B and T ALL leukemic stem cells (LICs) and mechanisms adopted by these LICs.
    12. 3390/cells10051218 provides an excellent summary of the side effects of pro-oxidant on normal HSCs and suggests the use of natural components and pro-oxidant therapies tagged with a linker to be released in high ROS environments. However, something that is not discussed, yet we believe is one of the most important considerations of ROS in ALL, is the targeting of ROS production and in particular the role of NOXs in this pathology; which are topics that form a major focus of our discussion.

We trust that this summary highlights some of the major differences between our manuscript and that of 10.3390/cells10051218. We contend that such points of distinction are sufficient to justify the publication of our article on what is clearly an important field of investigation.

Round 2

Reviewer 2 Report

The authors respond sucsesfully to the comments

Reviewer 3 Report

The manuscript has been substantially improved therefore it is now suitable for publication